# Dendritic Cells and Their Crucial Role in Modulating Innate Lymphoid Cells for Treating and Preventing Infectious Diseases

**DOI:** 10.3390/pathogens14080794

**Published:** 2025-08-08

**Authors:** Yeganeh Mehrani, Solmaz Morovati, Fatemeh Keivan, Tahmineh Tajik, Diba Forouzanpour, Sina Shojaei, Byram W. Bridle, Khalil Karimi

**Affiliations:** 1Department of Pathobiology, Ontario Veterinary College, University of Guelph, Guelph, ON N1G 2W1, Canada; ymehrani@uoguelph.ca; 2Division of Biotechnology, Department of Pathobiology, School of Veterinary Medicine, Shiraz University, Shiraz 71557-13876, Iran; sulmaz.morovati@gmail.com; 3Department of Microbiology and Immunology, School of Veterinary Medicine, University of Tehran, Tehran 14179-35840, Iran; fatemeh.keivan@ut.ac.ir (F.K.); d.forouzanpour@ut.ac.ir (D.F.); sina.shojaei@ut.ac.ir (S.S.); 4Department of Pathobiology, School of Veterinary Medicine, Ferdowsi University of Mashhad, Mashhad 91779-48974, Iran; ta.tajik@mail.um.ac.ir

**Keywords:** dendritic cells, innate lymphoid cells, infectious diseases

## Abstract

Two key players in the immune system, dendritic cells (DCs) and innate lymphoid cells (ILCs), interact in a crucial way to fight infectious diseases. DCs play a key role in recognizing pathogens, and ILCs respond to cytokines released by DCs. This response triggers the production of specific effector cytokines that help control pathogens and maintain the body’s barrier integrity. DCs have various receptors, including Toll-like receptors (TLRs), that detect microbial components and trigger immune responses. Likewise, ILCs act as essential initial responders in the immune system in viral, bacterial, and parasitic infections. Successfully managing diseases caused by pathogens mainly depends on the combined actions of DCs and ILCs, which work to suppress and eliminate pathogens. DCs also play a crucial role in activating innate and adaptive immune cell subsets, including ILCs. Furthermore, the use of DCs in developing vaccines and immunotherapy for cancers, along with the dedication of many researchers to improve immune responses through DCs, has increased interest in the potential of DC therapies for treating and preventing infectious diseases. This review examines approaches that may enhance DC vaccines and boost anti-infection immune responses by fostering better interactions of DCs with ILCs.

## 1. Introduction

Dendritic cells (DCs) facilitate immunity or tolerance by capturing and presenting antigens to T cells and delivering immunomodulatory signals through cell-to-cell interactions and cytokine release [1]. DCs are crucial for establishing an effective adaptive immune response through interaction with innate immune components like natural killer cells [2]. Their applications in diagnostics and research, such as vaccine production and virus propagation, and their crucial involvement in various DC-based immunotherapies—especially in cancer treatment—make them important targets for disease prevention and therapeutic development [3].

Innate lymphoid cells (ILCs) represent a diverse group of cells primarily located in non-lymphoid peripheral tissues [4]. The functions of ILCs are varied, encompassing the regulation of tissue and metabolic homeostasis, defense against infectious diseases, and involvement in the pathology of chronic inflammation and cancers. ILCs are a crucial source of innate effector cytokines, such as IFN-γ, IL-4, IL-5, IL-9, IL-13, IL-17, and IL-22, which are released in response to mediators and cytokines produced by epithelial, stromal, or other immune cells [5]. Since their identification, ILCs have been recognized as the innate equivalent of T cells. ILCs and T cells share numerous characteristics, including common progenitors, mechanisms of transcriptional regulation, and secretion of effector cytokines [6].

ILCs are classified based on their expression of transcription factors and the production of effector cytokines. In contrast to T cells, ILCs quickly react to various environmental signals, such as alarmins, cytokines, neuropeptides, hormones, and eicosanoids [4]. Additionally, they could take on new functions and phenotypes in response to changing environmental signals through a mechanism referred to as functional plasticity. The tissue-resident characteristics of ILCs enable them to respond rapidly to specific stimuli in the tissue and coordinate innate and adaptive immune responses during infections [5].

While DCs have been widely studied as antigen-presenting cells in vaccine design, their role in directly modulating ILCs remains largely unexplored. Most existing DC-based vaccines have been designed to stimulate adaptive immune responses, particularly cytotoxic T cells, with limited consideration of early innate effectors like ILCs. This represents critical oversight, especially given the tissue-localized, rapid-response functions of ILCs during infection. Integrating ILC-directed modulation into DC vaccine strategies introduces approaches to enhance mucosal immunity and tissue-specific protection, outcomes where traditional vaccine platforms often fall short.

This review proposes a framework for engineering DC-based vaccines that purposefully engage ILC subsets. We argue that strategic manipulation of the DC–ILC axis can bridge innate and adaptive immunity in ways not currently achieved by conventional vaccine approaches.

## 2. DCs and Their Subsets

DCs are key antigen-presenting cells (APCs) that initiate and modulate immune responses by detecting pathogens, processing antigens, and activating T cells [6]. Derived from hematopoietic stem cells (HSCs) via myeloid progenitors, DC development is regulated by transcription factors (e.g., PU.1, IRF8, BATF3) and shaped by local cytokines such as fms-like tyrosine kinase 3 ligand (*Flt3l*), granulocyte-macrophage colony-stimulating factor (GM-CSF), and interleukin-4 (IL-4), which influence their differentiation into steady-state or inflammatory subsets [7,8,9,10] (Figure 1).

### 2.1. Functional Hallmarks of DCs

A defining feature of DCs is their dual role: efficient antigen uptake via phagocytosis, macropinocytosis, and receptor-mediated endocytosis, followed by antigen presentation on major histocompatibility complex (MHC) molecules to naïve T cells in lymphoid tissues [11]. Upon recognizing pathogen-associated molecular patterns (PAMPs) through pattern recognition receptors (PRRs) like TLRs, DCs undergo maturation, marked by increased expression of co-stimulatory molecules (CD80, CD86, CD40), upregulation of MHC I and II, and cytokine production (e.g., IL-12, IL-6, and type I interferons (IFNs)) [12,13,14]. This integration of antigen presentation, co-stimulation, and cytokine signaling enables DCs to activate and direct T cell differentiation into specific effector subsets.

### 2.2. Major Subsets of DCs

DCs are generally categorized into three main subtypes: conventional DCs (cDCs), plasmacytoid DCs (pDCs), and monocyte-derived DCs (mo-DCs). These subtypes differ in their developmental origin, tissue distribution, and immunological function [15] (Figure 1).

#### 2.2.1. Conventional Dendritic Cells

cDCs stimulate CD8^+^ and CD4^+^ T cells by presenting antigen via MHC class I and II molecules. They are classified into two main subsets: conventional DC type 1 (cDC1) and conventional DC type 2 (cDC2) [16].

cDC1s specialize in cross-presentation, delivering exogenous antigens on MHC class I to CD8^+^ cytotoxic T cells. They are key players in antiviral and antitumor immunity, helping to maintain tolerance to self-antigens [17]. Additionally, they produce IL-12, which promotes T helper 1 (Th1) differentiation and enhances the activity of cytotoxic T lymphocytes (CTLs) and natural killer (NK) cells [18].

cDC2s present antigens on MHC class II to CD4^+^ Th cells, thereby supporting the differentiation of Th2 and Th17 subsets. They contribute significantly to immune responses against extracellular pathogens and are important for mucosal immunity [19].

#### 2.2.2. Plasmacytoid DCs

pDCs are a unique subset of DCs that can produce large amounts of type I IFN (IFN-α and IFN-β) in response to viral nucleic acids. This response is primarily mediated by the endosomal TLR7 and TLR9 [20]. As a result, pDCs can rapidly generate antiviral defenses even without the involvement of adaptive immune responses [21]. pDCs play a role in regulating immune responses, possess strong IFN signaling capabilities, and are associated with certain autoimmune diseases [22]. There is an ongoing debate about whether pDCs are a subset of DCs or a distinct lineage within the DC family, given their significant differences in development and function [23].

#### 2.2.3. Monocyte-Derived DCs

Mo-DCs are generated in response to inflammation, infection, or tissue injury [24]. Mo-DCs are the most commonly used DC subtype for in vitro immunological modeling [25]. A specific study by Paris et al. (2022) [25] demonstrated that human Mo-DCs, when infected with adenoviruses, can activate autologous ILCs in vitro. The co-culture resulted in the upregulation of activation markers (e.g., CD69) and the secretion of cytokines, including IFN-γ and IL-13. This activation varied based on the virus type and the presence of neutralizing antibodies, supporting the use of Mo-DCs for modeling ILC interactions during infection [25].
Figure 1Development and Subsets of Dendritic Cells (DCs). The developmental pathway of DCs from HSCs in the bone marrow is shown. Under the influence of cytokines, such as *Flt3l*, GM-CSF, and IL-4, as well as key transcription factors, HSCs give rise to various DC subsets, including cDCs, moDCs, and pDCs. These subsets differ in origin and function, but all play crucial roles in antigen presentation, immune activation, and cytokine production [6,7,8,11,15,26].
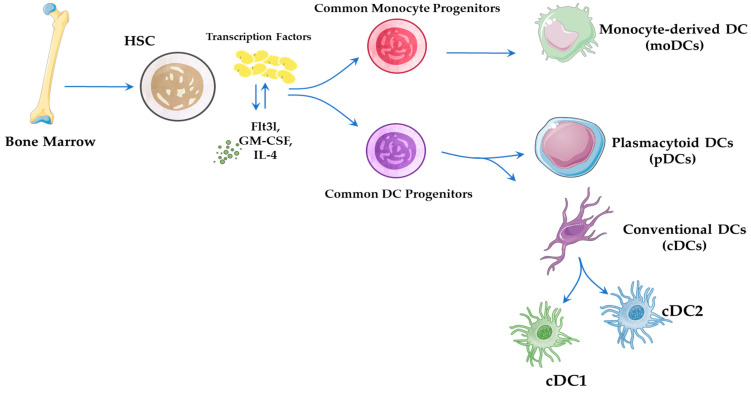


#### 2.2.4. Relevance to Infectious Diseases and Immunotherapy

The functional diversity of DC subsets shapes immune responses in infectious diseases, with each subset contributing to pathogen recognition and immune activation [27]. In DC-based therapies, selecting the appropriate subset, such as cDC1s, is critical for efficacy [21]. Enhancing antigen presentation or limiting autoimmune activation (e.g., through modulation of pDC) enables more precise and personalized immunotherapy strategies. Modulating pDC activity enables better control of immune responses tailored to the patient’s condition. By enhancing or suppressing type I interferon production, we can boost antiviral defense or reduce harmful inflammation. This flexibility supports more targeted and personalized immunotherapy, improving both safety and effectiveness in infectious disease treatment [21].

## 3. Innate Lymphoid Cells (ILCs) and Their Subsets

ILCs are tissue-resident lymphocytes primarily found at barrier sites, such as the intestine and lung [28]. Lacking rearranged antigen receptors, they act as the innate counterparts of T cells and respond rapidly to local signals by producing cytokines, such as IFN-γ, IL-4, IL-5, IL-9, IL-13, IL-17, and IL-22 [29]. ILCs contribute to tissue homeostasis and early immune defense (Figure 2).

ILCs are categorized into three primary groups based on the specific cytokines they produce, their phenotypic characteristics, and their developmental pathways [29]. Functionally, ILC1s, ILC2s, and ILC3s parallel CD4^+^ Th1, Th2, and Th17 cells, respectively, based on their cytokine profiles and roles in immune responses, while NK cells function similarly to CD8^+^ cytotoxic T cells [28].

### 3.1. ILC1s

Group 1 innate lymphoid cells (ILC1s) and NK cells collectively comprise the broader category of Group 1 ILCs. NK cells are the prototypical group 1 ILC, but ILC1s are a more recently identified group with diverse subtypes. These ILC1s represent the innate counterparts to Th1 cells and are activated in response to intracellular pathogens, such as viruses and tumors. They secrete interferon-gamma (IFN-γ) upon stimulation with cytokines IL-12, IL-15, and IL-18 [30]. ILC1 development and function depend on the transcription factor T-bet [31,32].

### 3.2. ILC2

ILC2s release type-2 cytokines, including IL-4, IL-5, IL-9, IL-13, and amphiregulin, in response to TSLP, IL-25, and IL-33 [30]. They play a role in the innate immune defense against large extracellular parasites and allergens [28]. IL-13 and amphiregulin contribute to the repair of tissue damage caused by helminthic and viral infections [33,34]. ILC2s are dependent on the transcription factors GATA3 and RORα for their function [30]. Furthermore, GATA3 is crucial for both the maintenance of the ILC2 population in vivo and their effector functions [35].

### 3.3. ILC3

Group 3 ILCs include natural cytotoxicity receptor (NCR)^−^ILC3s, NCR^+^ ILC3s, and LTi cells, all of which rely on the transcription factor RORγτ for their function and can secrete IL-17, IL-22, or both simultaneously [28]. They play a role in combating bacterial infections by secreting IL-22, which prompts epithelial cells to produce antimicrobial peptides [36].
Figure 2Different ILC subsets, their main cytokines, and primary roles in infectious disease defense. ILC1s release interferon-gamma (IFN-γ) in response to cytokines such as IL-12, IL-15, and IL-18. IFN-γ activates infected cells, including macrophages (MCs), promoting the production of nitric oxide and reactive oxygen species that are essential for controlling parasites. Both cell types require the transcription factor T-bet for differentiation and are activated in response to intracellular pathogens [1,30]. ILC2s, acting as the innate counterparts to Th2 cells, produce type-2 cytokines, such as IL-4, IL-5, IL-9, IL-13, and amphiregulin, when stimulated by TSLP, IL-25, and IL-33. The differentiation of ILC2s requires the transcription factors GATA3 and RORα, which are activated in response to helminths and environmental agents [30,37]. ILC3s produce IL-22 and IL-17 when stimulated by IL-23 and IL-1β, and they represent the innate equivalents of Th17 cells. The differentiation of ILC3s requires the transcription factor RORγτ, and they become activated in response to extracellular bacteria and fungi [28,37].
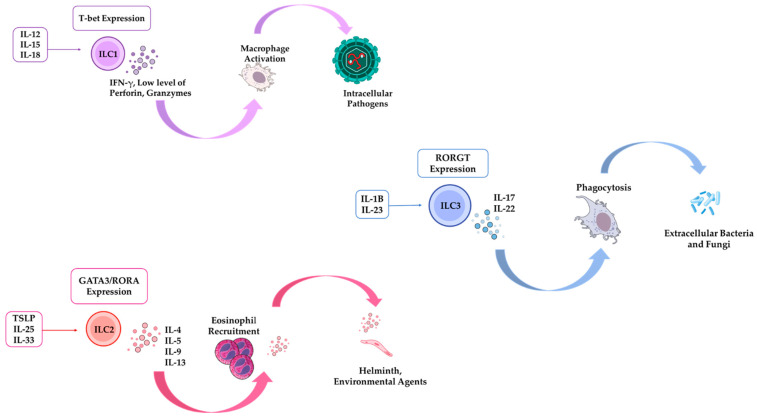


### 3.4. ILC Plasticity and Trans-Differentiation

ILC subsets display plasticity and can alter their phenotype and function in response to polarizing signals within the local tissue, provided they express the necessary cytokine receptors and key transcription factors [38]. Experimental evidence from fate-mapping and adoptive transfer studies in mice has shown that specific gut-resident ILC3 subsets can differentiate into IFN-γ-producing ILC1s, a process associated with downregulation of RORγt and upregulation of T-bet [28,39,40]. Cytokines, such as IL-1B, IL-15, and IL-12, appear to drive this shift and have been similarly shown to induce human IL-22-producing ILC3s and ILC2s to acquire an ILC1-like, IFN-γ-producing phenotype in vitro [38,41]. Recent findings suggest that ILC2s can also transition into ILC1s both in vitro and in vivo [41].

This plasticity contributes to cellular heterogeneity within tissues, which may be crucial for effective immune responses but also linked to the pathogenesis of inflammatory diseases such as Crohn’s disease and COPD [38,41]. Despite the increasing recognition of their adaptability, the full functional implications of ILC plasticity remain unclear and continue to be actively investigated [28].

ILCs play significant roles in tissue homeostasis and immunopathology through their interactions with DCs and effector cytokine networks [29,42]. Defining their specific contributions is critical to deepening our understanding of immune regulation and guiding the development of ILC-targeted therapies for infectious and inflammatory diseases.

## 4. DCs in the Context of Infectious Diseases

Beyond their essential role in antigen presentation, DCs dynamically interact with ILCs during infection. ILCs support DC maturation through cytokine and contact-dependent signals, while mature DCs, in turn, enhance ILC activity, forming a feedback loop that fine-tunes immune responses [43]. In lymphoid organs, chemokine signals direct the localization of DC subsets, shaping distinct immune programs. For example, cDC1s are found in the central T cell zone (TCZ), guided by CCR7 and chemokines CCL19/CCL21, and promote type 1 immunity via IL-12 [3,44] while cDC2s, expressing CXC chemokine receptor 5 (CXCR5), localize to the T–B cell border and are involved in Th2 and Th17 polarization [45]. This localization may also affect their interactions with ILCs during inflammation [3]. pDCs, key players in antiviral defense, secrete type I and III interferons in response to viral nucleic acids; however, their chronic stimulation leads to their functional exhaustion [20]. mo-DCs arise from monocytes during infection and produce pro-inflammatory cytokines, such as IL-6, IL-1β, TNF, IL-12, and IL-23, thereby activating ILC1s and ILC3s [46].

### Immune Evasion Through Impairment of DCs During Infectious Diseases

While DC activation is essential for initiating immune responses, many pathogens impair DC function to evade immune surveillance, thereby weakening host defenses. Pathogens often manipulate host signaling pathways to induce DC apoptosis, thereby undermining early immune activation [47]. For example, oral infection of mice with *Salmonella enterica* serovar Typhimurium induces the in vivo death of CD8-expressing cDC1s via MyD88 and Tumor necrosis factor receptor 1 (TNFR1) signaling, having a negative impact on the initiation of antibacterial immunity [48].

Even with properly developed DCs, dysregulation of DC migration may lead to abnormal positioning and improper activation of other immune cells, resulting in an imbalance of immune responses and even immune pathologies [49]. In acute SARS-CoV-2 infection, the expression of CD80, CD86, CCR7, and human leukocyte antigen–DR isotype (HLA-DR) after stimulation with TLR ligands was reduced in pDCs, cDC1s, and cDC2s, indicating potentially reduced maturation and migration of DC subsets [50]. Furthermore, in severe COVID-19 infections, the bronchoalveolar lavage fluid (BALF) displayed higher proportions of macrophages and neutrophils, but lower proportions of mDCs, pDCs, and T cells compared to moderate infections [50].

In HIV infection, DCs are depleted and functionally impaired, contributing to chronic immune dysfunction and viral persistence [49].

Under normal conditions, DCs shape the function and localization of ILCs through cytokine signaling and interactions with surface molecules. When DCs are dysregulated or depleted, this cross-talk is impaired, potentially leading to altered ILC activation patterns, reduced IFN-γ production by ILC1s, or a shift toward type 2 inflammation through ILC2s [51].

These examples demonstrate how pathogens disrupt the balance of DCs through apoptosis, impaired maturation, and altered trafficking. This ultimately weakens the immune defenses and impacts ILC dynamics. The disruptions not only compromise overall immune regulation but also interfere with the crucial communication between DCs and ILCs, a key interaction explored in the next section. From a therapeutic standpoint, administering DCs that have been sensitized outside the body may counter these evasion strategies by restoring targeted immune activation, including ILC engagement, and boosting the body’s response to pathogen clearance.

## 5. ILCs in the Context of Infectious Diseases

ILCs, as early responders in mucosal and barrier tissues, play a critical role in the initial immune response during infection [52]. Strategically located in sites, such as the gut, lungs, and oral mucosa, ILCs respond rapidly to microbial invasion before the activation of adaptive immunity [53,54]. Every ILC subset is activated by cytokines released from DCs and epithelial cells. In response, each subset produces distinct effector cytokines that help control pathogens and maintain the barrier integrity.

ILC1s and NK cells respond primarily to DC-derived cytokines including IL-12, IL-15, and IL-18 [55]. Upon activation, they produce TNF and IFN-γ, which enhance macrophage activity and provide protection against intracellular bacteria and viruses. TNF also contributes to macrophage-mediated bacterial clearance [56].

In helminth infections, epithelial tuft cells release IL-25 and IL-33, which indirectly influence DC function and promote ILC2 activation. ILC2s secrete IL-4, IL-5, IL-9, and IL-13, driving goblet cell hyperplasia and the recruitment of eosinophils and macrophages, key elements of the antihelminth response [56]. Since ILC2s produce IL-4, their activation may help suppress DC responsiveness to IFN type I, a mechanism previously demonstrated for IL-4 in the context of impaired DC-driven Th1 polarization [57].

ILC3s are key mediators of mucosal immunity against extracellular bacteria and fungi. DCs and macrophages secrete IL-23 and IL-1β in response to these pathogens, leading to the activation of ILC3s and the production of IL-17, IL-22, and GM-CSF [54,56]. IL-22 enhances barrier defense by inducing antimicrobial peptides, strengthening epithelial junctional integrity, and promoting epithelial fucosylation, mechanisms critical for controlling pathogen colonization [58]. IL-17 facilitates neutrophil recruitment to infected tissues, while GM-CSF supports the function of DCs and macrophages [56,59]. Notably, ILC3s can also present antigen via MHC class II and regulate T cell responses to commensal microbes, further integrating innate and adaptive immunity (Table 1) (Figure 3).

Through these distinct but complementary pathways, ILCs function as essential orchestrators of early immune responses. Therefore, understanding their role in infection provides valuable insight into developing strategies that boost mucosal defense and control pathogenic threats during the initial stages of disease.
Figure 3Interactions between DC and ILC subsets coordinate pathogen-specific immune responses. The figure illustrates how DCs engage with distinct ILC subsets to coordinate immune responses against specific pathogens. DC-derived cytokines activate different ILC subsets. IL-12 from DCs activates ILC1, inducing the production of IFN-γ and TNF, which contribute to defense against intracellular bacteria such as *Listeria monocytogenes* [55,60]. During influenza, epithelial damage and DC sensing of viral components both lead to the release of IL-33, which activates lung ILC2s. Upon activation, ILC2s proliferate and secrete IL-4 and amphiregulin. IL-4 suppresses DC responsiveness to type I IFN and reduces their expression of IL-12 and co-stimulatory molecules, impairing their ability to drive Th1 differentiation. Concurrently, ILC2-derived amphiregulin contributes to epithelial repair and mucosal healing, particularly during the resolution phase of infection [34,61]. IL-1β and IL-23 stimulate ILC3 to produce IL-17 and IL-22, enhancing mucosal immunity and supporting Th17 responses against extracellular pathogens, including fungi [62,63]. These interactions underscore the pivotal role of DC–ILC cross-talk in orchestrating pathogen-specific innate immune responses.
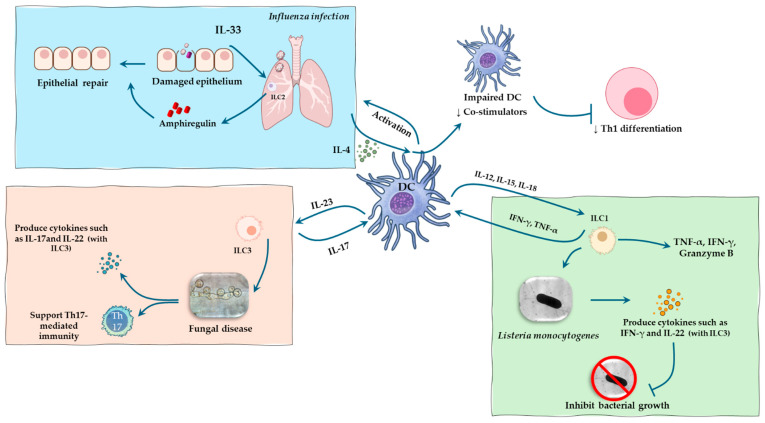

pathogens-14-00794-t001_Table 1Table 1ILC subsets respond to bacterial, viral, and fungal infections, underscoring their crucial role in the immune defense against a wide range of pathogens.PathogenILCs SubsetsFunctionsRef.Bacteria*Listeria monocytogenes*ILC1ILC3-Production of IFN-γ and IL-22 to inhibit bacterial growth and propagation[60]*Salmonella typhimurium*ILC3-Production of IFN-γ-Activating mucosal immunity (because of IL-22 and IL-17 secretion)[64,65]*Helicobacter pylori*ILC2-Production of IL-5 and promoting IgA production by B cells in stomach tissue[66,67]ViralInfluenzaILC2-Regulate inflammation and restore the integrity of epithelial airway in response to IL-33-Promote lung homeostasis by regulation of IL-5 that raises the accumulation of eosinophils[34,68]CytomegalovirusILC1-IFN-γ production and TRAIL expression regulate antiviral immunity by promoting cytotoxicity and limiting Cytomegalovirus (CMV) replication.[69,70]Fungal*C. albicans*,*A. fumigatus*ILC3-Respond to Aspergillus spp. via TLR2 → IL-2 → IL-22-Support Th17-mediated immunity[62,63]

### 5.1. ILCs and Bacterial Infections

ILCs contribute to mucosal immunity and antibacterial defense at barrier sites [71]. While their roles in responding to pathogens, such as *Mycobacterium tuberculosis*, *Listeria monocytogenes*, *Salmonella typhimurium*, and *Helicobacter pylori* are well-documented, the modulation of ILC activity by DCs remains a critical area of investigation.

A key regulatory pathway involves cytokines derived from DCs that influence the function of the ILC subsets. DCs produce IL-23 and IL-1β, which activate ILC3s to secrete IL-17 and IL-22, supporting epithelial barrier function and antimicrobial peptide expression. This interaction is essential in infections, such as tuberculosis, where ILC3-driven IL-22 promotes early lung immunity and the formation of inducible bronchus-associated lymphoid tissue (iBALT) [58].

During bacterial gastroenteritis, DCs activate ILC3s via IL-23 signaling to support host defense against *S. typhimurium*. However, *S. typhimurium* exploits this pathway by inducing IL-23 through TLR5-Myd88 signaling in DCs, leading to elevated IL-22 production from ILC3s, which in turn facilitates infection. Deletion of ILC3s, but not T cells, enhances bacterial clearance. The pathogen can also invade ILC3s directly and induce caspase-1–mediated pyroptosis. Casp1 deficiency results in increased ILC3 survival and IL-22 production, which correlates with a higher bacterial burden. These findings suggest that regulating ILC3 activity is essential to limit *S. typhimurium* replication and IL-22–mediated pathogenic effects [72].

AHR signaling, which integrates environmental and microbial signals, modulates interactions between DCs and ILC3s. AHR expression in both cell types promotes IL-22 production and enhances GM-CSF-dependent support of intestinal macrophages. ILC3s play a pivotal role in gut immune homeostasis by protecting the intestinal mucosa against various pathogens [73,74]. They mediate this function primarily through the secretion of IL-22, IL-17, and GM-CSF, which stimulate epithelial cells to produce antimicrobial peptides, such as RegIIIβ and RegIIIγ [75]. Additionally, ILC3s regulate T cell responses to commensal microbes via MHC class II expression, support tolerogenic DC function through GM-CSF, and modulate epithelial glycosylation, collectively contributing to mucosal barrier integrity and immune balance [76,77]. Mice lacking *Ahr* exhibit reduced ILC3 numbers and impaired antibacterial responses, underscoring the significance of DC-ILC3 cross-talk in mucosal defense [31].

Understanding the regulation of ILCs by DCs across bacterial infections may reveal novel therapeutic strategies for enhancing mucosal immunity and limiting inflammation.

### 5.2. ILCs and Viral Infections

Distinct ILC subsets reside in mucosal tissues and mount subset-specific responses to diverse pathogens [78]. During viral infections, STAT1 signaling regulates the balance of ILC subsets, promoting antiviral IFN-γ^+^ ILC1s while suppressing potentially immunopathologic IL-5^+^ and IL-13^+^ ILC2s and IL-17A^+^ ILC3s. This regulation occurs through both ILC-intrinsic and ILC-extrinsic mechanisms, with STAT1 controlling the expression of IL-33 and IL-23, which promote ILC2 and ILC3 responses, respectively [79].

ILC1s and NK cells respond rapidly during early phases of viral infections to effectively suppress viral replication at the site of infection [80]. Tissue-resident ILC1s provide early host protection through the rapid production of IFN-γ, which is stimulated by IL-12 produced by cDC1s that express XCR1 [55].

In severe COVID-19, the disease is characterized by elevated levels of immunosuppressive cytokines like transforming growth factor-beta (TGF-β) and IL-10, which dampen the cytotoxic functions of NK cells and ILC1s [81]. Although the precise DC-derived cytokines involved remain under investigation, it is known that cDC1s produce IL-12 during early SARS-CoV-2 infection, which can activate ILC1s to produce IFN-γ [55]. This helps control viral replication and limit inflammation. Reduced ILC2 numbers in severe COVID-19 [82] suggest that impaired DC–ILC2 signaling, potentially through IL-33 or IL-7, may disrupt tissue repair and epithelial homeostasis.

During Cytomegalovirus (CMV) infection, tissue-resident ILC1s are similarly activated early and contribute to viral control by producing IFN-γ and expressing TNF-related apoptosis-inducing ligand (TRAIL). This process is driven by cDC1-derived IL-12, which primes ILC1s to produce IFN-γ, suppressing viral replication at infection sites [55]. Moreover, evidence from mouse models suggests that liver-resident ILC1s acquire memory-like features after re-exposure to CMV, a process linked to DC-mediated cytokine signaling. These memory ILC1s upregulate IL-18R and are promptly activated upon repeated pathogen encounters [83,84], showing that DC conditioning can contribute to long-term innate immune memory.

In influenza infection, epithelial cell damage leads to the release of alarmins, such as IL-33, which activate lung-resident ILC2s [32]. DCs, particularly those sensing viral components via TLRs, amplify this effect by releasing IL-33 [61]. This cytokine supports ILC2 proliferation and amphiregulin production, which promotes epithelial repair [85]. While direct DC-ILC2 cross-talk during influenza remains to be fully characterized, the functional relationship between them contributes to resolving inflammation and promoting mucosal healing.

ILC3s, while not directly infected by HIV or simian immunodeficiency virus (SIV), are substantially depleted during infection due to the pro-inflammatory environment. pDCs infiltrate mucosal tissues and produce high levels of IFN-α, upregulating CD95 on ILC3s and promoting apoptosis [86]. Additionally, chronic DC-derived cytokine exposure (e.g., IL-12, IL-15) suppresses RORγt expression in ILC3s, disrupting mucosal immunity [87,88]. Emerging therapeutic strategies, including IL-7 supplementation or use of vaccine adjuvants to enhance ILC3 populations, have shown potential in restoring mucosal homeostasis and immune balance in chronic viral infections.

Collectively, these observations highlight the crucial roles of ILC1s and ILC2s in antiviral defense, where they contribute through rapid cytokine production, direct cytotoxicity, and support for tissue repair [89]. DCs shape ILC responses via a spectrum of cytokines, including IL-12, IL-18, IL-33, and IFN-α, establishing a dynamic DC-ILC axis that is pivotal for the early control of viral replication, the regulation of infections, such as SARS-CoV-2, CMV, influenza, and HIV/SIV, and the maintenance of mucosal homeostasis [54].

In conclusion, the interplay between ILCs, DCs, and cytokine signaling networks is fundamental to orchestrating effective mucosal immune responses against a wide range of viral pathogens. These insights reinforce the therapeutic value of modulating the DC–ILC axis to optimize antiviral immunity and tissue integrity during infection.

### 5.3. ILCs and Fungal Infections

Fungal infections are a significant health risk, particularly for immuno-compromised individuals. Experimental studies suggest that ILCs play a crucial role in initiating the immune response to fungi by primarily secreting IL-17 [90].

*Candida albicans* is a major opportunistic fungal pathogen in humans, capable of causing both mucosal and systemic infections, particularly in the skin, oral cavity, and vaginal tract [91]. In vivo studies using murine models have underscored the essential role of ILCs in mediating fungal clearance at oropharyngeal mucosa. This effect is largely mediated through IL-17 production in response to IL-23 signaling [90]. Additionally, exposure of the respiratory tract to *C. albicans* stimulates the recruitment and activation of various immune cells, notably NK cells, macrophages, DCs, and ILCs. These cells contribute to a protective, neutrophil-independent immune response primarily through IL-22-driven upregulation of antimicrobial peptides [92]. Aspergillus infections primarily affect the lungs and lower respiratory tract, leading to severe disorders such as invasive pulmonary aspergillosis and allergic bronchopulmonary aspergillosis. Research shows that ILCs, via TLR2, enhance the production of IL-2 and IL-22, which increase susceptibility to Aspergillus spp. allergic airway responses [93]. Additionally, ILC3-secreted IL-22 has been shown to possess protective effects on lung function upon exposure to acute *A. fumigatus* infection [62].

Furthermore, ILC3 imbalance is associated with an increased susceptibility to fungal infections [94]. In aged mice, ILC3 dysregulation was characterized by an increase in NKp46^+^ ILC3 and a decrease in CCR6^+^ ILC3, notably the CCR6^+^CD4^+^ subset. This led to a reduction in the secretion of IL-22 and IL-17A, impairing antifungal immunity. ILC3s also regulate Th17 cell activity, which is crucial for maintaining mucosal immune integrity [63]. In combination, these findings reveal the multifunctionality of ILCs in orchestrating antifungal immunity and suggest their potential use as therapeutic tools to modulate host defense.

Understanding ILC biology, including their development and interaction with DCs, is key to advancing infectious disease therapies.

Coordinated research efforts will be essential to translate these insights into effective ILC-based therapies for fungal infections.

## 6. DC-ILC Cross-Talk in Infectious Diseases

ILCs, DCs, and T cells collectively constitute three distinct functional immune modules and engage collaboratively to initiate a targeted immune response. The innate immune response involves various innate cell types, including neutrophils, monocytes, as well as ILCs and DCs [95]. Different subsets of DCs and ILCs contribute uniquely to immune regulation, helping tailor adaptive responses to the nature of the invading pathogen [96]. They contribute to the primary functions of the three main types of immune responses. Type I immunity is responsible for combating intracellular pathogens, including viruses and specific bacteria. Type II immunity provides defense against helminths and allergens. Conversely, Type III immunity primarily targets extracellular bacteria and fungi [37].

Understanding DC-ILC interactions may provide a potential strategy for therapeutic interventions aimed at enhancing immunity against pathogens, restoring immune homeostasis, and improving outcomes in infectious diseases.

### 6.1. Type I Immunity (Intracellular Microbes)

Type I immunity is essential for protecting against intracellular pathogens and can be divided into two main responses [27,35]: (a) a cytotoxic response involving NK cells, pDCs, and CD8^+^ T cells. NK cells and pDCs are among the initial responders to viral infections. NK cells lyse infected cells via granules containing perforins and granzymes and are activated by proinflammatory cytokines, such as IL-12, IL-15, IL-18, and type I IFNs, mainly produced by DCs [97]. pDCs secrete large amounts of type I IFN, which is critical for antiviral defense, and help activate NK cells through cytokine production [37,98] (Figure 4).

IL-15 plays a vital role in the development and activation of NK cells, trans-presented via the IL-15Rα-chain expressed by DCs [37,99]. The absence of pDCs during the early response to murine cytomegalovirus (MCMV) infection results in decreased production of type I IFN and reduced activation of NK cells [37,100].

(b) Intracellular defense mechanism maintained by ILC1s, cDC1s, and Th1 cells. Both ILC1s and cDC1s deliver inflammatory signals that activate Th1 cells, thereby enhancing the immune response against intracellular pathogens, such as *Toxoplasma gondii* [37]. Although both NK cells and ILC1s produce IFN-γ, ILC1s respond more rapidly than NK cells when exposed to various viruses, including MCMV, Sendai virus (SeV), and Puerto Rico/8 (PR8) influenza virus [51,101]. Activated CD8α^+^ cDCs release IL-12, which triggers IFN-γ secretion from ILC1/NK cells. IFN-γ further activates infected cells, such as macrophages (MCs), promoting the production of nitric oxide and reactive oxygen species, which are essential for controlling parasites. Simultaneously, activated CD8α^+^ cDCs travel to lymph nodes, facilitating the differentiation of Th1 cells that improve parasite clearance. The Batf3-dependent CD8α^+^ cDC is vital for initiating type I responses to *Toxoplasma gondii* infection, as it uniquely secretes IL-12, which is necessary for NK cell activation and subsequent IFN-γ production [1].

ILC1s protect against Sendai virus and influenza infections by producing high levels of IFN-γ in experimental settings [102]. Both ILC1s and NK cells contribute to combating intracellular pathogens, such as *T. gondii*, *L. monocytogenes*, *Salmonella typhimurium*, and viruses, through the synthesis of IFN-γ [51]. In mouse models of *Clostridium difficile* and *Toxoplasma gondii* infection, ILC1s contributed to host defense through the secretion of IFN-γ and TNF. Mechanistically, ILC1s are stimulated by IL-12 from cDCs to produce IFN-γ [97].

### 6.2. Type II Immunity (Against Helminths and Environmental Substances)

Type II immunity defends against helminths and environmental agents, triggered by ILC2, a subset of cDC2, in interaction with Th2 cells [37]. Both ILC2s and Klf4-dependent cDC2 play vital roles as innate cells in this immune response, while Th2 cells offer adaptive support. This immunity is influenced by the cytokines IL-5, IL-9, and IL-13, with ILC2s being the primary innate producers of IL-5 and IL-13. These cytokines promote IL-4 production by mast cells (MCs), which is essential for Th2 differentiation. Additionally, IL-13 from ILC2s stimulates CCL17 production by lung and dermal cDC2s, which helps attract memory Th2 cells in response to allergens [37]. ILC2-derived IL-13 is also crucial for facilitating the migration of activated lung DCs into draining lymph nodes, where they prime naive T cells to differentiate into Th2 cells [9] (Figure 4). ILC2s express MHC class II and can activate T cells (though less effectively than DCs), leading to IL-2 production that promotes ILC2 proliferation and the secretion of Th2-associated cytokines that aid in worm expulsion [51]. These interactions between cDC2s and ILC2s are critical in orchestrating Type II immune responses during parasitic and allergic infections. Understanding how DCs shape ILC2 function provides a foundation for developing targeted therapies that modulate type II immunity in infectious disease settings.
Figure 4Type II immunity (response to helminths and environmental agents). Upon helminth infection, epithelial cells release alarmins that activate ILC2s. Activated ILC2s secrete IL-5 and IL-13. IL-5 recruits eosinophils, which contribute to parasite killing, while IL-13 promotes cDC2 migration to draining lymph nodes by enhancing their response to CCR7 ligands [101]. In the lymph nodes, cDC2s present antigens and express OX40L to support naive CD4^+^ T cell priming and Th2 differentiation. Th2 cells, in turn, produce IL-2, which feeds back to enhance ILC2 proliferation and survival [9,51,103].
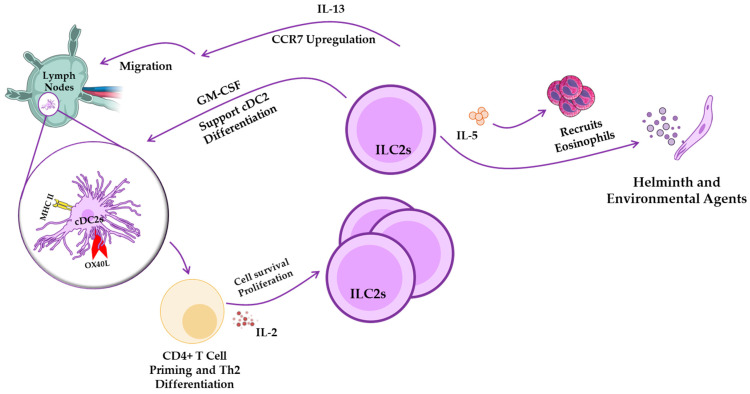


### 6.3. Type III Immunity (Against Extracellular Bacteria and Fungi)

Finally, Type III immunity plays a critical role in defending extracellular bacteria and fungi, involving ILC3s, a specific subset of cDC2s, and Th17 cells. The presence of ILC3s and Notch2-dependent cDC2s is essential for an effective immune response against extracellular pathogens and fungi (Figure 5). IL-23 is crucial for the secretion of IL-22 by ILC3s; thus, cDC2s modulate intestinal type III immunity through their interactions with both Th17 and ILC3 cells [37].

In response to inflammatory signals, DCs and epithelial cells produce IL-1β and IL-23, which promote the elevated production of IL-22, along with other inflammatory mediators such as IL-8, GM-CSF, IL-26, and TNF. These factors influence neutrophils, dendritic cells, and epithelial cells to ensure optimal protection against pathogens [104].

Emerging evidence suggests that ILC3-derived GM-CSF may indirectly support DC and regulatory T cell (Treg) function in the gut. While this contributes primarily to homeostasis, disruption of this axis could have implications for the host–microbe balance during infection [51].

ILC3s are actively involved in immune responses against *Citrobacter rodentium*, *Clostridium difficile*, *Salmonella enterica*, *Listeria monocytogenes*, and *Toxoplasma gondii* [105]. IL-22 is vital for eliciting a protective immune response against the attaching and effacing bacterium *Citrobacter rodentium* in mice [104]. Infection with *C. rodentium* activates CD11b^+^ cDCs in the small intestine through an uncharacterized mechanism. Once activated, these tissue-resident CD11b^+^ cDCs produce elevated levels of IL-23, which in turn, stimulates IL-22 production in RORγt-dependent ILC3s, potentially influencing other cell types as well. IL-22 prompts the epithelial layer to secrete the bactericidal lectin RegIIIγ, which is crucial for clearing *C. rodentium*. Additionally, CD11b^+^ cDCs migrate to the mesenteric lymph node, where they facilitate the differentiation of T cells into Th17 and Th22 subsets. These effector T cells return to the infection site and serve as an adaptive source of IL-22. It should be noted that the secretion of IL-23 by tissue-resident CD11b^+^ cDCs is enough to trigger IL-22 production in ILC3s, thereby ensuring effective local control of *C. rodentium* [1].

The ability of DCs to regulate ILC3 activity through cytokine-driven mechanisms, particularly via IL-23 and IL-22, emphasizes their essential role in orchestrating protective Type III immune responses against extracellular pathogens.

## 7. DC-Based Vaccines and Modulation of ILCs: Insights from Cancer Studies

DC-based vaccines function by priming DCs with tumor antigens, thereby enhancing immune recognition and elimination of malignant cells. Beyond stimulating adaptive immunity, DC vaccines may also shape innate responses by influencing ILC phenotypes and functions toward antitumor roles [106,107,108].

ILCs exhibit functional plasticity and respond dynamically to local tissue, inflammation, and immune signals [108]. By engineering DCs to secrete cytokines, such as IL-12 and IFN-γ, vaccines can skew ILCs toward an ILC1-like phenotype, which is associated with IFN-γ production and enhanced cytotoxic immune activity [109]. In our recent mouse melanoma model, we observed that DC vaccination led to an increase in ILC2 frequencies in the spleen, draining lymph nodes, and lungs [109,110]. Moreover, these ILC2s exhibited altered cytokine profiles, suggesting that the vaccine drove modulation of ILC2s. We also demonstrated that DC vaccines increased both NCR^+^ and NCR^−^ subsets of ILC3s, and the secretion of IL-17 and IL-22 increased post-vaccination [108]. These findings suggest that DC vaccines may be used to combat infectious diseases by engaging the ILCs.

## 8. DC-Based Vaccines and Modulation of ILCs in Infectious Diseases

Infectious diseases are unpredictable and can lead to outbreaks or pandemics. Historically, vaccination has played a pivotal role in disease control and eradication, as exemplified by the eradication of smallpox. The ongoing search for effective vaccines has driven the development of new platforms that elicit stronger immune responses and allow scalable production, surpassing traditional live attenuated or inactivated vaccines [110,111].

A hallmark of systemic infections and sepsis is DC depletion, which correlates with higher mortality in septic shock [47]. Furthermore, DCs are skilled at antigen cross-presentation (i.e., presenting to both CD4+ and CD8+ T cells) and play a vital role in connecting innate and adaptive immunity against invading pathogens. In this context, various strategies have been explored for utilizing DCs as vaccine platforms [112]. Originally developed to stimulate antigen-specific T cell responses, DC-based vaccines are now also recognized for their role in modulating innate immunity, especially through interactions with ILCs [107]. The interplay among DC biology, vaccine development, and ILC modulation offers significant potential for treating and preventing infectious diseases, where the effective coordination of innate and adaptive responses is crucial for pathogen clearance and long-term immunity [6].

Emerging research [108] suggests that DC vaccines can modulate ILC in infectious scenarios, potentially enhancing type-specific immune responses through cytokines such as IL-12 and IL-23. This capacity to modulate ILC function offers new opportunities for enhancing vaccine-induced protection beyond T cell activation.

### 8.1. Overview of DC Vaccines in Infectious Disease

One of the most widely studied approaches is the ex vivo DC vaccine. In this process, autologous DCs, typically derived from peripheral blood monocytes, are isolated, cultured, and differentiated in the laboratory. They are loaded with pathogen-derived antigens and matured using immune stimulants. After reinfusion into the patient, the activated DCs migrate to lymphoid tissues and stimulate T cells [113]. This approach has enhanced T cell responses in diseases such as HIV, hepatitis C (HCV), and tuberculosis, although the clinical impact remains modest [114,115,116,117,118]. However, the complex manufacturing process and scalability issues continue to pose significant challenges [113].

To overcome logistical and immunological limitations of ex vivo DC vaccines, in situ vaccination strategies have gained attention. They use DC activators (e.g., TLR agonists, Flt3L) to recruit and mature endogenous DCs at the site of administration [112]. In preclinical models (e.g., influenza, HIV), this method improves delivery and tissue targeting [114,119]. However, precise immune modulation is crucial to preventing the induction of tolerance [112].

An important category of vaccines is mRNA-loaded DC vaccines. In this approach, DCs are transfected with messenger RNA that encodes microbial antigens. The translated antigen is presented via MHC I and II pathways [120,121]. This technology has garnered attention for its ability to elicit strong cytotoxic T cell responses and its rapid adaptability in targeting new pathogens, as demonstrated in experimental vaccines for SARS-CoV-2 [122]. Challenges include poor delivery, premature activation, and complex manufacturing [120].

DNA-loaded DC vaccines utilize plasmid DNA that encodes pathogen antigens to induce the DCs to express these antigens endogenously. These vaccines can stimulate strong cellular immune responses, particularly by activating cytotoxic T lymphocytes through cross-presentation pathways [123]. Though tested in diseases like malaria, their lower efficiency limits clinical impact [124].

In summary, DC-based vaccines for infectious diseases involve various technologies, including ex vivo-generated DCs, in situ targeting systems, and DCs loaded with RNA or DNA. Each approach presents its own strengths and trade-offs in terms of immunogenicity, practicality, and clinical applicability. The ongoing optimization and integration of these platforms show promise for enhancing protection against challenging infectious diseases.

### 8.2. Strategies to Modulate ILC Activity Through DC-Based Vaccines

Given the crucial role of ILCs in initiating early immune responses, maintaining tissue integrity, and coordinating inflammation [125], DC-based vaccines present a valuable opportunity to modulate and direct ILC activity with specificity. DC-based vaccines, which have traditionally focused on adaptive immunity, can be strategically targeted to modulate ILC activity. This enhances innate responses alongside T cell activation, thereby improving the scope and efficacy of vaccine-induced protection [126] (Figure 6).

#### 8.2.1. Subset-Specific Targeting of DCs

Distinct subsets of DCs preferentially interact with specific populations of ILCs because of their unique cytokine secretion profiles and receptor expression patterns. For example, targeting cDC1s with delivery systems or ligands that engage surface receptors, such as XCR1 or C-type lectin domain-containing 9A (CLEC9A), can enhance ILC1 responses. This enhancement is characterized by the production of IFN-γ, which plays a critical role in defending against viral infections and intracellular bacteria [127].

Directing antigens to cDC2s enhances the activation of ILC3s through cytokines like IL-23 and IL-1β. These cytokines support ILC-mediated antiviral responses and immune regulation [74].

When selectively activated, pDCs produce significant amounts of type I IFN, particularly via TLR7 or TLR9. These interferons indirectly influence the function of ILCs in antiviral immunity and play a role in immune regulation [133].

Moreover, mo-DCs can be specifically tailored to influence various subsets of ILCs, including ILC1, ILC2, and ILC3, through the production of cytokines such as IL-12, IL-33, and IL-23 [134].

LCs, specialized DCs in the epidermis that express langerin, are thought to influence local ILC2 and ILC3 activity by modulating the epithelial cytokine environment [135]. Although not fully defined, LC–ILC interactions may shape local immunity, especially after intradermal vaccination (ID) [136].

By customizing antigen delivery systems for specific DC subsets, vaccines can generate specialized ILC responses that are adapted to the type of pathogen and tissue, thus improving both the accuracy and effectiveness of immune protection (Figure 7).

#### 8.2.2. Cytokine Engineering of DCs

DCs can produce specific cytokines that influence the activity of innate lymphocytes. This interaction enables the precise modulation of the cytokine environment, promoting the expansion and functional polarization of targeted ILC subsets [140,141]. Tissue-resident cDC1 rapidly produces IL-12 following viral infection, driving early IFN-γ production by ILC1s in a STAT4-dependent manner, which limits viral replication and enhances host protection [55]. Additionally, DC-derived IL-23 and IL-1β boost ILC3 activity. These cytokines support epithelial barrier defense through the production of IL-17 and IL-22 [137]. Similarly, the release of IL-33 or TSLP from engineered DCs activates ILC2 responses [138]. These findings highlight the utility of cytokine-engineered DCs in orchestrating innate ILC responses to enhance immune defense during infection. A notable example of cytokine engineering in DCs involves modifying DC progenitors to co-express interleukin-12 (IL-12) and FLT3L. In a recent study, these engineered DCs were shown to differentiate in vivo and induce potent antitumor immune responses without the need for antigen loading. The co-expression of IL-12 and FLT3L enhanced T cell activation, NK cell recruitment, and tumor regression in mouse models [142]. This approach demonstrates the therapeutic potential of cytokine-programmed DCs for antigen-agnostic cancer immunotherapy.

#### 8.2.3. Spatiotemporal Optimization of Vaccine Delivery

The location and timing of interactions between DCs and tissue-resident ILCs are crucial for effective immune modulation. Mucosal delivery methods, such as intranasal or oral administration, enhance DC–ILC communication at barrier sites, where specific ILC subsets, including ILC1s in the lungs, ILC2s in mucosal tissues, and ILC3s in the gut, are strategically positioned to rapidly respond to infection. These routes mimic natural infection pathways and ensure immune activation in situ [131,139]. Targeting these routes allows vaccines to enhance local cytokine signaling, activate innate defenses, and improve protection at mucosal surfaces.

#### 8.2.4. Rational Adjuvant Selection

The choice of vaccine adjuvants is crucial in determining the response of DCs and, in turn, how this affects group 1 and group 3 of ILCs. Agonists of PRRs, such as TLR ligands (e.g., CpG, poly(I:C)), STING activators, and RIG-I ligands, promote DC maturation and shape cytokine profiles that influence ILC subsets [129]. For instance, TLR9 agonists may promote the activation of ILC1 through the production of IL-12, while nucleotide-binding oligomerization domain (NOD2) ligands may encourage the expansion of ILC3 by increasing IL-23 production [129,130,143]. This highlights the role of adjuvants as modulators of innate immunity via DC–ILC interactions. Incorporating ILC-targeted strategies into DC-based vaccine platforms may optimize both early innate and long-term adaptive immune protection.

### 8.3. Strategies for Utilizing DCs as Vaccine Platforms in Infectious Diseases

#### 8.3.1. Ex Vivo Peptide-Loaded DC

In these procedures, autologous DCs are obtained from the patient either by isolating them from peripheral blood or by differentiating them in vitro from monocytes or CD34^+^ hematopoietic precursor cells mobilized from the bone marrow. Culturing purified monocytes with GM-CSF and IL-4 yields numerous mo-DCs [110]. Due to the limited number of peripheral blood DCs and the complexity of generation protocols, CD34^+^ hematopoietic precursor cell-derived DCs have been used in only a few cancer-related trials, while mo-DCs remain preferred in most clinical applications [49].

To load obtained DCs with antigens, they can be delivered through peptide pulsing, viral vectors (e.g., adenovirus-vectored TB vaccine [144]), or mRNA transfection (e.g., mRNA encoding HIV Tat, Rev, and Nef proteins [145]). Cytokines, CD40L, and TLR agonists are used to mature DCs before reinfusion into the patient [110,146]. This ex vivo approach has demonstrated efficacy in murine models and clinical trials targeting parasitic diseases (e.g., visceral leishmaniasis), fungal infections (e.g., *Candida albicans*, *Cryptococcus gattii*), and viral infections (e.g., Influenza, Herpes simplex virus (HSV), HIV) [110].

#### 8.3.2. In Vivo Targeting of DC

In this strategy, a monoclonal antibody (mAb), genetically engineered to target a specific DC surface receptor, or a ligand, is combined with an antigen (or encoding nucleic acid) and delivered in vivo to target DCs [110]. Various DC receptors have been evaluated for vaccine targeting, including Fc receptors (FcR), CD11c, LOX1, CD40, DCIR, or C-type lectin receptors such as DEC-205 (CD205), DC-SIGN (CD209), mannose receptor (CD206), Langerin (CD207), or DNGR1/Clec9A, or XCR1. CD40 and DEC-205 are among the most widely used targets in cancer and infectious disease vaccines. One advantage of this strategy is that DCs are not manipulated ex vivo, avoiding changes in activation phenotype. However, some studies indicate that targeting DEC-205 without an adjuvant induces tolerance rather than immunity [110]. Another option is to use nanoparticles that contain either the chosen antigen or the DNA/mRNA encoding that antigen [110,147]. Taken together, these strategies illustrate the adaptability of DC-based vaccine platforms in shaping ILC responses to improve infectious disease outcomes. By utilizing subset-specific DC targeting, cytokine engineering, spatial delivery routes, and rational adjuvant design, vaccines can be tailored to induce tissue-specific and pathogen-appropriate ILC responses. Importantly, integrating ILC modulation into DC-based vaccine design not only enhances early immune activation but also establishes a more balanced and durable immune response, thereby bridging innate and adaptive immunity more effectively. Future studies should prioritize translating these DC–ILC interactions into clinically viable vaccine platforms, particularly for infections where mucosal immunity, rapid ILC activation, or immune recovery following sepsis are critical.

## 9. Approaches to Enhance DC Vaccines and Boost Anti-Infection Immune Responses by Fostering Better Interaction with ILCs

DC-based vaccines have mostly been studied for cancer treatment [148]. While preclinical studies often yield promising results, clinical translation remains challenging [149]. Understanding how DCs interact with ILCs can inform the design of vaccines that promote more effective immune responses. Influenza-infected DCs enhance NK cell activation, increasing their cytolytic activity and IFN-γ production. In turn, NK cells modulate DC function and influence the activation and cytokine production of T cells during viral infections [150]. Enhancing DC-NK cell interactions through cytokine modulation or the use of targeted adjuvants may promote stronger and longer-lasting anti-infective immunity. Thus, fostering productive DC-ILC engagement is a promising strategy for the development of the next generation of DC-based vaccines.

IL-13, primarily produced by ILC2s, influences the activity of DCs and other ILC subsets during the early phases of immune activation. This cytokine plays a modulatory role in shaping the downstream adaptive immune response by promoting type 2 immunity and regulating local inflammation. IL-13Rα2 may signal via STAT3 under certain conditions [151]. It helps regulate IL-13 signaling in ILC2s and DCs.

Targeting this axis affects IL-13/IFN-γ balance and immune polarization [152]. Strategically targeting this signaling pathway may enhance the efficacy of DC-based vaccines by promoting the recruitment and activation of specific DC and ILC subsets, leading to stronger and more precisely directed protective immunity [153].

Intracellular infections are challenging to treat because pathogens reside within the host cells, where they are protected from many immune mechanisms and drugs. Current therapies often fail to eliminate these infections. To address this, DC-based immunotherapies have been developed. In one approach, DCs pulsed with *Leishmania donovani* antigens were transferred into mice. This increased IFN-γ production and activated cell-mediated immunity. To further enhance this response, DCs were genetically engineered to secrete IL-12. Mice that received these modified DCs exhibited even stronger immune responses and significantly reduced parasite burden [128,154].

Liver-resident ILC1s play a regulatory role in the activation of CD103^+^ DCs during viral infections. Specifically, blocking the inhibitory receptor NKG2A on ILC1s enhanced the production of IFN-γ, a cytokine that promotes the activation of DCs. As a result, activated DCs more effectively stimulate CD8^+^ T-cell responses [155]. This mechanism highlights a potential strategy for enhancing DC-based vaccines by modulating the ILC1 signaling pathways.

Effective vaccines rely on strong antigen presentation by DCs. NKp46 ILCs have been shown to suppress CD8^+^ T-cell responses by targeting DCs. They do so via NKG2D-dependent killing, which reduces the number of functional antigen-presenting DCs available to activate CD8^+^ T cells [156]. These findings suggest that modulating the interaction between ILCs and DCs, for example by inhibiting suppressive ILC activity, could enhance the efficacy of DC-based vaccines.

Both the route of vaccine administration and the choice of viral vector significantly influenced the recruitment and activation of ILCs and DCs at the vaccination site. For example, intranasal vaccination induces ST2/IL-33R^+^ ILC2s, whereas intramuscular (IM) vaccination leads to the induction of IL-25R^+^ and TSLPR^+^ ILC2s subsets. Moreover, the type of viral vector used can modulate the immune response. Recombinant Fowlpox Virus is associated with both cellular and humoral immunity and is characterized by high NKp46^+^ ILC1/ILC3-derived IFN-γ, low ILC2-derived IL-13, low IL-17A levels, and increased recruitment of CD11b^+^ CD103^−^ cDCs. In contrast, recombinant Modified Vaccinia Ankara and Influenza A vectors, which are associated with low-avidity T cell responses, induced higher levels of ILC2-derived IL-13, distinct cytokine profiles, and enhanced recruitment of cross-presenting DCs. Additionally, the intranasal delivery of Rhinovirus and Adenovirus type 5 vectors resulted in elevated IFN-γ production by NKp46^+^ ILC1/ILC3 and ILC2-derived IL-13, which recruited CD11b^−^ B220^+^ pDCs. Considering that pDCs play a role in antibody differentiation, this cytokine environment suggests the potential for effective humoral immunity [126]. Fan et al. (2019) [132] showed that ID immunization with inactivated EV71/CA16 antigens activated DCs and ILCs more strongly than IM delivery. Early ILC–DC colocalization suggested ILCs support DC-driven adaptive immunity [132] (Figure 7).

Despite decades of progress in DC-based vaccine development, the integration of ILCs into these platforms remains largely overlooked. This represents a critical gap, as ILCs are rapid responders that operate at barrier sites and play essential roles in early pathogen control, inflammation shaping, and tissue repair. Most current vaccine strategies focus on T cell responses, underutilizing the unique potential of ILCs as programmable effectors.

This work proposes a new immunological framework in which DCs are not only antigen-presenting cells but also strategic modulators of ILC activity. By harnessing DC subset-specific targeting, cytokine engineering, spatial delivery routes, and adjuvant selection, vaccines can be designed to fine-tune ILC responses in a tissue- and pathogen-specific profile. Such approaches open the door to more localized, durable, and balanced immunity, especially in infectious diseases where current vaccine strategies fall short.

The current review positions ILCs modulation as a core design principle in DC-based vaccine development for infectious diseases, a shift in perspective that has the potential to redefine how we build next-generation immunotherapies by aligning DC precision with ILCs to orchestrate smarter, more effective immune protection from the ground up.

## 10. Conclusions

DCs are central to the coordination of immune responses, bridging innate and adaptive immunity through their potent antigen-presenting functions and cytokine-producing capacity. In recent years, it has become increasingly clear that their role extends beyond T cell priming to include the modulation of ILCs, a diverse group of tissue-resident effectors that play a critical role in early host defense, inflammation, tissue repair, and immune regulation. The interaction between DCs and ILCs is emerging as a key regulator of immune responses to infectious diseases, with significant implications for both treatment and prevention strategies. DCs influence ILCs activation and polarization in a subset-specific and tissue-specific manner. For instance, cDC1s enhance ILC1 and NK cell-mediated cytotoxicity via IL-12 and type I interferons, while cDC2s and mo-DCs promote ILC2 and ILC3 activity through IL-33, IL-1β, and IL-23. These interactions are not unidirectional; ILCs also modulate DC function, forming a dynamic feedback circuit that shapes immune responses based on local tissue cues and the type of pathogens. This bidirectional cross-talk establishes a functional bridge that connects early innate immunity with downstream adaptive memory, allowing for rapid, site-specific, and robust responses to infection.

Infectious pathogens, particularly those targeting mucosal tissues, frequently modulate the innate-adaptive interface to evade immune clearance. As such, the interaction between DCs and ILCs represents a strategic target for vaccine design and immunotherapy. DC-based vaccines, originally developed to stimulate antigen-specific T cells, are now being explored for their ability to engage ILCs and enhance innate responses. Preclinical and clinical data suggest that engineered DCs, programmed with specific cytokine profiles or loaded with mRNA and antigens, can be used to selectively activate subsets of ILCs. These strategies can augment mucosal immunity, drive tissue-specific protection, and potentially induce innate memory-like responses.

Technological advances, such as single-cell and spatial transcriptomics, nanoparticle delivery systems, and biomaterial-based platforms (e.g., hydrogels) now enable more precise manipulation of DC-ILCs interactions in vivo. These innovations enable targeted antigen presentation, controlled cytokine release, and spatiotemporal regulation of immune activation, paving the way for next-generation vaccines that coordinate both arms of immunity. Moreover, the possibility that ILCs may retain memory-like functions suggests that modulating this axis could yield durable protection complementary to traditional adaptive memory.

However, several challenges must be addressed before these strategies can be widely translated into clinical practice. Most mechanistic insights have been derived from murine models or simplified in vitro systems, which do not fully capture the heterogeneity of human immune tissues. Interspecies differences in DCs and subsets of ILCs, limited access to human mucosal samples, and the complexity of engineering DCs under GMP conditions all present real barriers. Furthermore, the immunosuppressive strategies employed by many pathogens—such as disrupting DC maturation or subverting DC–ILCs signaling—highlight the need for vaccines that not only engage but also protect this critical immune interface.

In conclusion, the DC–ILC axis represents a paradigm shift in our understanding of immune regulation during infection. Integrating ILC-directed strategies into DC-based vaccines and therapies broadens the immunological reach of these platforms, enabling faster, more localized, and longer-lasting protection. By leveraging the spatial precision, cytokine plasticity, and reciprocal communication of DC–ILCs circuits, we can move beyond traditional vaccine approaches toward precision immunotherapies that reflect the full complexity of host–pathogen interactions. As the threat of emerging and re-emerging infectious diseases grows, targeting this axis offers a timely and promising avenue for both prophylactic and therapeutic intervention.

## Figures and Tables

**Figure 5 pathogens-14-00794-f005:**
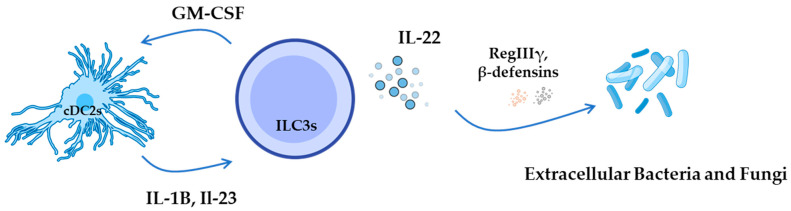
Type III immunity against extracellular bacteria and fungi. ILC3s are activated by IL-1β and IL-23 secreted by cDC2s in response to microbial stimuli. Upon activation, ILC3s produce IL-22, which acts on epithelial cells to induce the expression of antimicrobial peptides, such as RegIIIγ and β-defensins, thereby enhancing barrier defense. Additionally, ILC3-derived GM-CSF promotes cDC2 differentiation and supports the maintenance of local DC populations, reinforcing mucosal immunity [37,104].

**Figure 6 pathogens-14-00794-f006:**
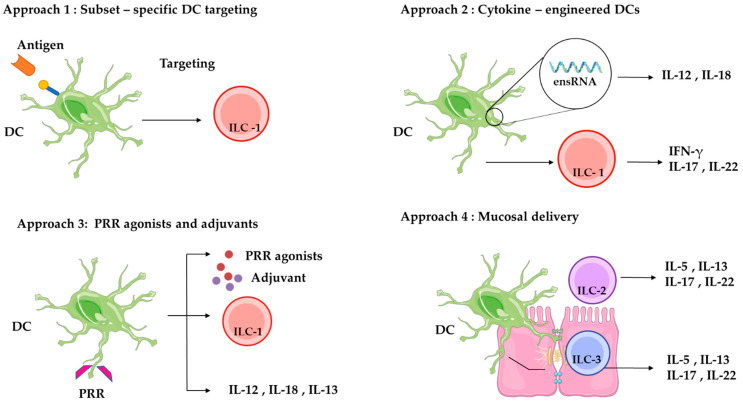
Strategies to modulate ILCs via dendritic cell-based vaccines. This diagram summarizes four main approaches for modulating ILC responses using DC-based vaccines: (1) subset-specific DC targeting [127], (2) cytokine engineering of DCs [128], (3) use of PRR agonists and adjuvants [129,130], and (4) mucosal delivery [131,132]. Each strategy aims to fine-tune cytokine signaling to activate ILC1, ILC2, or ILC3 subsets, thereby enhancing immunity, mucosal protection, and tissue repair.

**Figure 7 pathogens-14-00794-f007:**
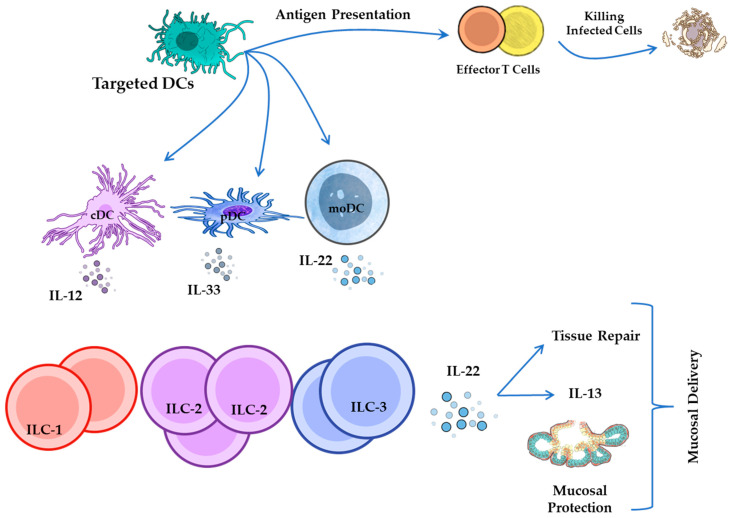
Targeting DC subsets to modulate ILC responses. This schematic illustrates how distinct DC subsets, including cDCs, pDCs, and moDCs, can be engineered to present antigens and secrete specific cytokines that selectively activate ILC subsets [15]. For instance, cDCs produce IL-12 to activate ILC1s, pDCs secrete IL-33 to stimulate ILC2s, and moDCs release IL-22 to support ILC3 function. These cytokine-driven interactions enhance mucosal protection and promote tissue repair. In parallel, antigen presentation by DCs also primes T cells, which differentiate into effector subsets that contribute to pathogen clearance [55,113,127,134,137,138,139].

## Data Availability

Not applicable.

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
