# Peer review of "Dendritic Cells and Their Crucial Role in Modulating Innate Lymphoid Cells for Treating and Preventing Infectious Diseases"

_pathogens, 2025, doi:10.3390/pathogens14080794_

Round 1
Reviewer 1 Report
Comments and Suggestions for Authors
The submitted review “The Role of Dendritic Cells in Modulating Innate 2 Lymphoid Cells Against Infectious Diseases: Exploring Dendritic Cell Therapies for the Treatment and Prevention of Infectious Diseases” by Yeganeh Mehrani et al., describes in great detail the different subsets of DCs and ILCs , their role and crosstalk in the context of infectious diseases . The review also examines in depth how DCs-based vaccines can be improved in order to properly modulate also ILCs and so enhance immune response to vaccines. This is an emerging and interesting issue to be discussed in scientific community.
This reviewer has no other suggestions and comments.
Author Response
Please see the attachment."

Reviewer 2 Report
Comments and Suggestions for Authors
Dear Authors,
First of all, congratulations for your interesting work. I hope that my hints will help you in the next steps of improvement and the final manuscript will be really valuable for the readers. In fact, I do not have many comments, since your revision is prepared in a very good and organised way, including excellent graphics and tables.
It might be a good idea to rethink the title, although it explains the paper well, it is not very encouraging, hence, it won’t attract many potential readers.
Moreover, gene names should be written in italics, in opposite to the protein names, according to the rules of genetic consensus. Please, familiarise yourself with the rules and change the manuscript accordingly. Examples of rules summary can be found on websites such as: https://www.gmb.org.br/geneprotein-nomenclature-guidelinesor https://academic.oup.com/molehr/pages/Gene_And_Protein_Nomenclature
Some sentences in the manuscript are highlighted on yellow - why? I think it is only an omission.
There are many parasites, bacteria etc, mentioned in the text - would you consider creating a table to summarise them?
Finally, I would like to thank you for the excellent figures and graphs you have prepared for the document, they enhance the value of your work and facilitate the understanding process.
Author Response
Please see the attachment."

Reviewer 3 Report
Comments and Suggestions for Authors
The article entitled “The Role of Dendritic Cells in Modulating Innate Lymphoid Cells Against Infectious Diseases: Exploring Dendritic Cell Therapies for the Treatment and Prevention of Infectious Diseases” is a review that explores the immune functions of dendritic cells (DCs) and innate lymphoid cells (ILCs), with an emphasis on their potential applicability in vaccine design and immunotherapies.
The authors have delivered a well-structured, comprehensive, and well-written review that provides a valuable and updated perspective on the interplay between DCs and ILCs in the context of infectious diseases. The topic is highly relevant, and the manuscript effectively integrates recent findings with conceptual developments in the field.
- I have only one suggestion for improvement:
When citing references older than 10 years, please ensure they are supported or corroborated by more recent studies to strengthen the manuscript's timeliness and scientific rigor.
Author Response
Please see the attachment."
